# Giant clams as open-source, scalable reef environmental biomonitors

**Daniel Killam** [1]*, **Diane Thompson** [2], **Katherine Morgan** [1], **Megan Russell** [1]

**1** Biosphere 2, University of Arizona, Oracle, AZ, United States of America, **2** Department of Geosciences, University of Arizona, Tucson, AZ, United States of America

* Daniel.E.Killam@gmail.com

## Abstract

Valvometry, the electronic measurement of bivalve shell opening and closing, has been demonstrated to be a valuable biomonitoring technique in previous ecological and environmental studies. Valvometric data has been shown to relate significantly to pollution, predation, animal stress and feeding activity. However, there is a need for valvometric techniques applicable to coral reef environments, which may provide critical insights into reef resilience to ocean warming and acidification. Giant clams are endemic to coral reefs and hold great promise as valvometric recorders of light availability, productivity and other environmental variables. Despite this promise, prior valvometric work on giant clams has been limited by specialized hardware less accessible to developing countries where many coral reefs are found. Here we report on an open-source approach that uses off-the-shelf components to monitor smooth giant clam (*Tridacna derasa*) valve opening behavior, and tests this approach in the simulated reef environment of the Biosphere 2 Ocean. Valvometric data corroborates the influence of light availability on diurnal behavior of giant clams. The clams basked during daylight hours to expose their photosymbionts to light, and adopted a partially-closed defensive posture at night. The animals showed variations in the frequency of complete closures, with most occurring during night-time hours when the animals prioritize filter-feeding activity, clapping their valves to expel pseudofeces from their gills. Closure frequency showed a significant relation to pH and a significant lagged relationship to chlorophyll-a productivity, which are both a function of algal productivity in the Biosphere 2 Ocean tank. These results suggest that the animals fed on phytoplankton following periodic bloom events in the Biosphere 2 Ocean during the experiment. We propose that giant clams exhibit behavioral plasticity between individuals and populations, and advocate for the more widespread use of valvometry to enable comparative studies of reef environment and animal health.

## Introduction

Giant clams (Tridacninae) are a subfamily of tropical coral reef-dwelling bivalves that reach unusual size through a partnership with symbiotic algae. Their dual reliance on autotrophy

**Data Availability Statement:** All data files are held in the University of Arizona Research Data Repository: 10.25422/azu.data.21642272 All relevant Arduino and R-based data processing code is hosted with the supplemental data on the

corresponding author's Github account at https://github.com/danielkillam/OpenValvometry.

**Funding:** DK- Brown Foundation, University of Arizona Postdoctoral Fellowship DT- University of Arizona RII The funders had no role in study design, data collection and analysis, decision to publish, or preparation of the manuscript.

**Competing interests:** The authors have declared that no competing interests exist.

and heterotrophy and indeterminate growth makes them long-living indicators of coral reef health. Researchers have long understood their potential as reef biomonitors, through the geochemical signals encoded in their shells [1], their absorption of microplastics and other contaminants [2], and in recent years, through valvometric analysis of their behavior [3, 4]. Valvometry is the study of bivalve shell opening and closing through proximity sensors such as electrodes [5], Hall effect magnetic sensors [6], machine-learning analysis of video [7], and accelerometers [8]. The technique has been applied to a growing assemblage of bivalve taxa to quantify and monitor their behavior through a high-frequency, non-invasive approach [9]. Bivalve shell movement can measure the presence of environmental contamination [10–13], predator avoidance [14], storm events [15], underwater noise [8], and other behavioral and environmental factors.

In giant clams, shell opening and closing have been found to be controlled by diurnal light availability [3], with the clams basking wide open to optimize photosynthetic production in the daylight hours, and closing partially at night. The clams close when exposed to predation [16], temperatures approaching their maximum thermal tolerance [3], and to minimize exposure to ultraviolet radiation, such as among intertidal populations [4]. Researchers in ethology have increasingly used giant clam shell closure as a model system to understand how mollusks balance food acquisition and predator avoidance [16]. Giant clams have eyes that can resolve object sizes [17, 18] and evaluate those objects relative to perceived threat [19]. The clams can habituate to repeated stimuli and prioritize energy expenditures towards reacting to novel or more severe threats [19]. They have also been found to change their valve closure behavior as they grow larger [20] and if exposed to distracting stimuli such as noise [21].

Giant clams are grown around the world for the reef aquarium and seafood industries [22]. These institutions are active in efforts to re-seed reefs with giant clams to increase wild populations [23]. Their stock could also potentially serve as distributed sources of valvometric data to better understand the behavioral ecology of giant clams in their native environments across the Indo-Pacific. However, shortcomings in previous approaches have precluded widespread application of valvometric monitoring of giant clams in aquaculture settings. First, past giant clam valvometric work has used specialized hardware such as electrodes calibrated using a specific proximity formula and connected to a waterproof wireless transmission box, which adds complexity in terms of preparation and deployment of sensors [3]. They have also used closed-source resources which could exclude the application of valvometry in the developing world, where most of the institutions practicing giant clam aquaculture are present. The use of open-source resources such as Raspberry Pi and Arduino would allow the broad selection of equivalent microcomputer and microcontroller devices to be deployed to replicate results, not necessitating the exact hardware used in the originating study. Further, past valvometric studies have often not provided comprehensive information on the sensors, installation procedures, loggers, and data storage that are needed to enable full reproducibility, resulting in researchers repeatedly reinventing the same procedures. Finally, as giant clams are a sensitive CITES-listed taxon, some of the approaches used for the attachment of valvometric sensors to common intertidal taxa like mussels or oysters are too harsh to use with giant clams, potentially disrupting their behavior in the process of application. Such methods have generally involved attaching small electrodes or Hall sensors directly to the shells of studied bivalves with hot glue or other air-curing adhesives [11]. These invasive measures could lead to exposure-induced mortality in giant clams or disrupt their behavior due to stress, particularly at the juvenile life stage when their stress tolerance is lower [24]. In general, the sensor attachment methods have not been reported in prior valvometric studies of giant clams [3, 4].

There is growing recognition of the value of open-source hardware such as Arduino [25] and Raspberry Pi [26] to counter these challenges and create reproducible, scalable research

instrumentation. Here we report on an inexpensive (<$300 USD), open-source, and noninvasive valvometric method which can be field-deployed in the tropics using off-the-shelf parts, providing the same data quality as previously reported methods. Critically, both Arduino and Raspberry Pi are widely-available, mass-market modules readily available in consumer markets worldwide. Further, we describe sensors with an attachment that cures underwater, minimizing the time the clams spend above water to less than ten minutes, but bonds weakly allowing easy detachment via the pry action of a screwdriver. Easy removal allows sensors to be replaced easily if repairs are needed, with minimal disruption to the clam. We demonstrate how such a system can be deployed and monitored to ensure maximum reliability over multi-month time periods, and provide code and guidelines to process, analyze and interpret the data in relationship to the clams' behavior and external environment. Using this approach, we report on giant clams' response to light availability, chlorophyll-a, phycoerythrin (a measure of cyanobacterial presence), DO/pH, and predation on diurnal and multi-week timescales in the simulated reef environment of the 2.5 million liter Biosphere 2 Ocean tank, where three juvenile individuals of *Tridacna derasa* were monitored using the system over a multi-month experiment.

## Methods

### The Biosphere 2 ocean as a valvometric testing laboratory

We obtained three individuals of *Tridacna derasa* from ORA Farms (Fort Pierce, Florida) on September 23, 2020 at around 5 months age. The clams grew over several months in 3 feet of water in the backreef lagoon of the 2.5 million liter ocean biome tank at Biosphere 2 in Oracle, Arizona [27]. Environmental data was obtained with several sensors. Light was sampled at 15-second intervals via a Li-Cor LI-192 planar Photosynthetically Active Radiation (PAR) sensor paired to an LI-1500 data logger. A planar rather than spherical sensor was selected as the symbionts receive most or all their insolation from above, channeled through rows of iridophores in their mantles [28]. The LI-192 is sensitive to wavelengths between 300–700 nm and measures photons/m$^2$s at ± 5% precision. The sensor was mounted on a Li-Cor 2009S lowering frame hanging 15 cm above the clams, close enough to capture light measurements representative of what the clams experienced, but far enough to avoid touching the clams when carried by a wave surge. Four Kessil AP9X LED aquarium lights were installed on a novel floating light rig 30 cm above the water's surface (S2 Fig), set to a standard "Tuna Blue" program to supplement the natural light coming through the glass of Biosphere 2 (Fig 2, S2 Fig). The baseline solar PAR received in the Biosphere 2 Ocean is only about 50% of the levels normally observed on natural reefs [27], which would not be sufficient to sustain the clams' symbiosis (which requires >200 µmol photons/m$^2$s in *T. derasa* [24]). A HOBO MX2501 pH logger and U26-001 Dissolved Oxygen (DO) Logger were hung from the lighting rig, collecting pH, DO, and temperature readings at 15-minute intervals. Around 30 m from the clams, a YSI Exo3 multiparameter sonde was hung from a boom to measure DO, chlorophyll-a (a marker of phytoplankton productivity) via a 6025 Chlorophyll Sensor, and Total Algal Phycoerythrin (TAL-PE, a measure of cyanobacterial productivity) via an EXO Total Algae PE Smart Sensor. Both data types were collected at 15-minute intervals in the form of Relative Fluorescent Units (RFUs). The clams were grown for three months with corresponding environmental and valvometric data logged throughout and all instrumentation calibrated on a weekly basis. All data is available in the Supplemental materials and the Dryad repository.

Temperatures were maintained at a consistent 25˚C, with diurnal variations of less than 0.25˚C, while pH values ranged from 8.1–8.25 pH units, also following a diurnal pattern controlled by light-mediated photosynthetic activity in the tank. Similarly, DO varied between

6.89 and 9.94 mg/L, also following a diurnal pattern with peaks during the daytime hours. Chlorophyll-a and phycoerythrin values showed high variability, with extreme peaks in July and August approaching the detection limit of the instruments and corresponding to visible increases in water turbidity.

Circulation in the Biosphere 2 ocean is provided by continuous pumps and a wave machine, simulating the water movement of a real-world barrier-reef and backreef-lagoon environments. The Biosphere 2 Ocean is host to a self-propagating community of plankton and benthic algae which represent a heterotrophic food source for the clams. The area where the clams are growing is covered with a loose pavement of coral/shell fragments, similar to the substrate that wild giant clams inhabit on coral reefs. In 9 months, the Biosphere 2 clams grew rapidly in height from ~ 3.5 cm to over 6 cm, in the range of growth rates observed from wild *T. derasa* in Palau [29]. The Biosphere 2 Ocean is thus a controlled setting that approximates a wild reef-lagoon environment in which to test the valvometric instrumentation.

## Valvometer construction

We obtained a series of electronic components to create a valvometric instrument, emphasizing open-source and/or inexpensive off the shelf options wherever possible. AsahiKasei EQ-730L analog Hall-effect magnetic sensors were obtained from GMW, Inc. for ~$1.67 per sensor. We soldered the sensors to the ends of 25-foot 4-ply braided copper cables in a vinyl casing, taking care to protect the sensors during soldering with a heat sink clip. We then covered the sensor ends of the cables with marine-grade dielectric grease (CRC Marine) and sealed them inside heat-shrink tubing. We covered the tubing with aquarium-grade silicone (Selsil) and left it to cure for 24 hours. The other end of the wiring was soldered to standard breadboard jumper cables. We inserted the voltage-in, ground, and analog-out jumper cables for each sensor into the 5 volt, ground, and analog-in ports of an Arduino UNO microcontroller, respectively, which provided voltage to the sensors and monitored returning voltage at a 5-second frequency from each sensor independently using its own analog-in port (Fig 1A). The Arduino was outfitted with a screw-terminal breakout module to keep the cables secured, while still allowing for easy removal during the prototyping phase. A DS3231 Real-Time Clock module was used for Arduino timekeeping. Finally, the Arduino was paired to a Raspberry Pi 4B computer via USB, which served as a data logger, a remote access client, and an independent battery power source for the Arduino (via a 3.7 V battery-based Uninterruptible Power Supply). Both were housed in a water resistant plastic housing. The Raspberry Pi logged the Arduino output as tab-delimited text files with 5 second frequency using the serial-logging function of the open-source, secure-shell client PuTTY.

Both Arduino and Raspberry Pi can individually serve as a valvometric voltage source, multiplexer and recorder, but we chose to use both combined, playing to their relative strengths. Arduino has extensive resources and ports available for logging analog sensor output on a continuous basis. We could have additionally used the Arduino for data logging and storage with an attached microSD-card module, but instead we elected to use a Raspberry Pi 4 computer for that purpose, as it allowed for remote access via a Virtual Network Client, as well as live onsite viewing of valvometric data on a monitor.

The Arduino was programmed via the Arduino integrated development environment using open-source libraries available from the Arduino website. The current time and three sensor voltage signals were programmed to export at 5-second intervals. The resulting files were then processed, merged, and analyzed via an R script executed in Rstudio. All code is available in the supplemental materials, the Dryad repository, and on the Github account associated with the authors.

## Valvometer installation and monitoring

The three juvenile clams were removed from the water individually to attach sensors, taking care to transport clams without removing their byssal threads from the shelly substrate to which they were attached. Care was taken to ensure they were only kept out of the water for a maximum of ten minutes. While giant clams are often cultured in the intertidal zone [30] and can survive hours of exposure, often with shells open [31], we sought to minimize their stress by limiting their time exposed to air. Once out of the water, their shells were wiped dry with a paper towel and a set of three stacked neodymium magnets that were coated in clear nail polish for waterproofing were attached to one valve with bSi IC-GEL Coral Frag Glue, chosen for its improved hold on partially wet surfaces. The silicone-tipped sensors were attached to the other shell valve using Star Brite underwater epoxy putty, chosen due to its limited production of volatiles during curing and its ability to cure underwater (Fig 1B). The adhesion of the putty was supplemented with frag glue, as the combined action of the two adhesives proved more effective at binding the silicone-coated sensor to the shell than either adhesive was alone. Sensors and magnets were attached at the broadest point of the valves near the umbo, with the cable leading to the posterior or anterior edges of the shell to prevent contact with the clams' sensitive upward-facing siphonal mantles (Fig 1B). After 4–5 minutes air curing, the clams were returned to their habitat. The clams were observed to reopen their shells within a few minutes of sensor attachment.

The sensors were monitored daily by investigating the live readout on the PuTTY client. During the prototyping process, some sensors suffered seawater intrusion and shorting. Saltwater intrusion was easily observable in the raw voltages being measured by the Arduino, which deviated from an average of around 2500 mV to either nearly 5000 mV or zero mV. Once aquarium-grade silicone was used instead of standard grade silicone, saltwater intrusion was no longer a problem. Sensor failure due to shorting resulted from failed soldering welds and/or pinching together of sensor contacts during installation. Such sensor failures were visible as a complete loss of diurnal signal in the clam's valvometric data, or a lack of record of known points when the clams were observed to be closed. These issues were remedied using high-temperature electrical tape wrapped around the sensor contacts to prevent them from

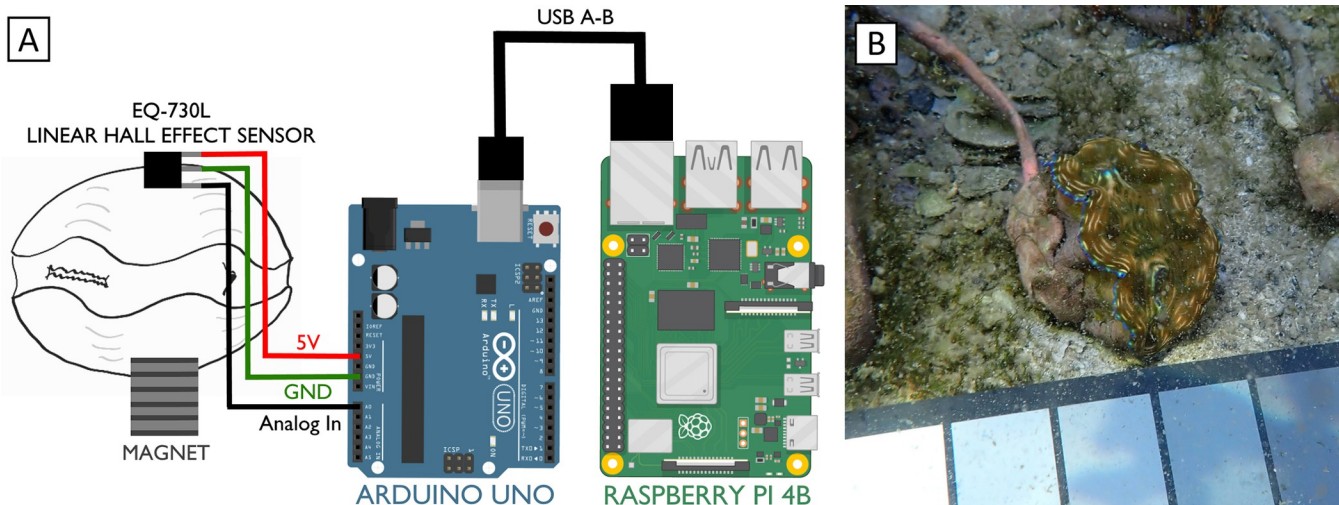

**Fig 1.** A): Schematic showing major components of the Hall effect sensing system, including the Hall effect sensor, magnet, Arduino UNO, and Raspberry Pi. Accessories not pictured: DS3231 RTC module (for Arduino timekeeping), Arduino screw terminal shield (for splitting 5V/ground ports across 4 additional sensors), and 3.7V Raspberry Pi UPS (for battery backup). B): A close-up view of a *Tridacna derasa* specimen with sensor attached.

touching. Once these two major sources of failure were addressed in early May 2021, the sensors operated through a three-month interval without any issues. A detailed step-by-step guide for sensor fabrication is available in the Supplemental Materials.

Valvometric sensor information was converted from raw mV values to a standardized daily z-score with the equation:

$$Z_t = \frac{mV_t - \mu}{SD}$$

, where $mV_t$ refers to the live-measured sensor voltage at time t, μ was the daily mean value in mV, and SD refers to the daily standard deviation in mV. Daily z-score values were used to control for day-to-day drift displayed by some sensors, which may have related to the "walking" movement of neighboring clams in relation to each other [16] allowing magnetic interference to register in their respective Hall effect sensors. Percent closure was calculated with the formula:

$$\% \ Closure_t = \left( \frac{Z_t - Z_{daily \ min}}{Z_{daily \ max} - Z_{daily \ min}} \right) * 100$$

Intermittent rapid closure events were delineated using an R function that identified local maxima/minima from the sensor z-scores in a 3-point (>15 second) moving window [32]. We used a three-point window as it aligned well with the behavior of the clams, which were observed to close briefly in response to researcher presence or filter-feeding activity and reopen within around fifteen seconds or longer. A shorter window (one or two points of heightened sensor z-score values) would risk false positive closure events which could be the result of sensor noise, and a longer window risked the clam reopening fully before a closure was measured by the sensor and delineated by the script. These closure events were then subset and analyzed in terms of frequency and length in seconds before reopening. Because all variables were non-normal as determined by Shapiro-Wilks tests and Q-Q plots, we related closures per day to daily mean light, chlorophyll-a, phycoerythrin, pH, and DO using a generalized additive model (GAM) [33]. GAMs allow for predictive analysis of interacting data of mixed distribution. The predictor variables are transformed through localized smooth functions, most commonly splines, which are combined additively to best explain the variance of the response variable (including proportional penalty functions based on the degree of smoothing of each variable) [34]. Critically, GAMs make no assumption of linearity in in the response variable. GAMs can therefore fit a response variable such as giant clam closure count per day in a nonlinear fashion relative to nonparametrically distributed variables such as pH. These characteristics make GAMs more relevant for analysis of behavioral data, where exponential relationships and threshold effects violate the base assumptions of traditional linear regression. Additionally, mixed effects are supported, allowing for the inclusion of predictor variables that are intercorrelated.

We additionally conducted lead/lag Cross Correlation Function (CCF) analyses relating the valvometric data to environmental data using the R function "ccf". Finally, to visualize variation in the periodicity of valvometric data across the 3 month monitoring period, we applied a Morelet wavelet transform with the R package "WaveletComp" [35]. Wavelet transformation allows for heat maps to be generated visualizing the strength and persistence through time of periodic dynamics in the data [36]. We were interested in the extent to which diurnal, tidal and supratidal periodicities may have influenced the clams' behavior. The data was averaged to minute-scale resolution to reduce computation time. The package also includes tools to reconstruct variability at certain periodicities; we reconstructed the 24-hour variability and

subtracted it out to obtain residuals, which were used to assess intra- and inter-diurnal periodicities in the valvometric data. Finally, we used wavelet coherence analysis between datasets to investigate the strength and phase of the relationship between valvometric data and pH, DO and light in the Biosphere 2 Ocean at 15-minute resolution (across periodicities over the duration of the experiment).

## Results

All three giant clams in this study showed patterns of opening in the hours leading to sunrise, typically around 3–4 AM (Fig 2A, S3 Fig). Peak light exposure occurred between 8–10 AM, when light levels regularly approached 600 μmol photons/m$^2$s. The LED lights were

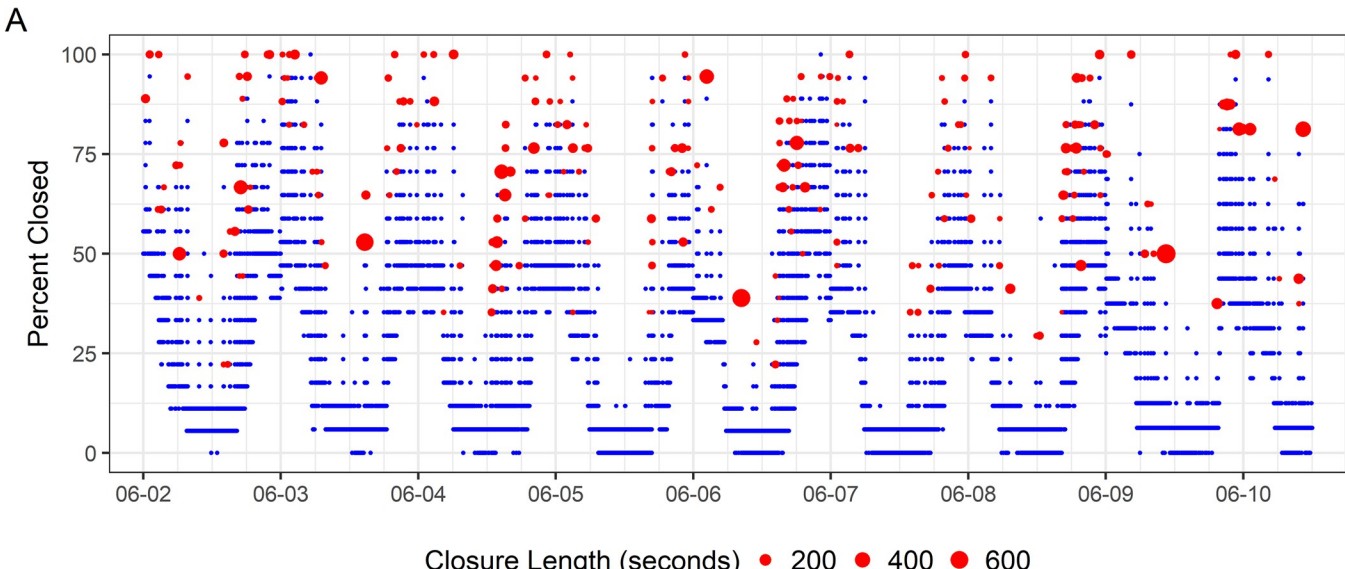

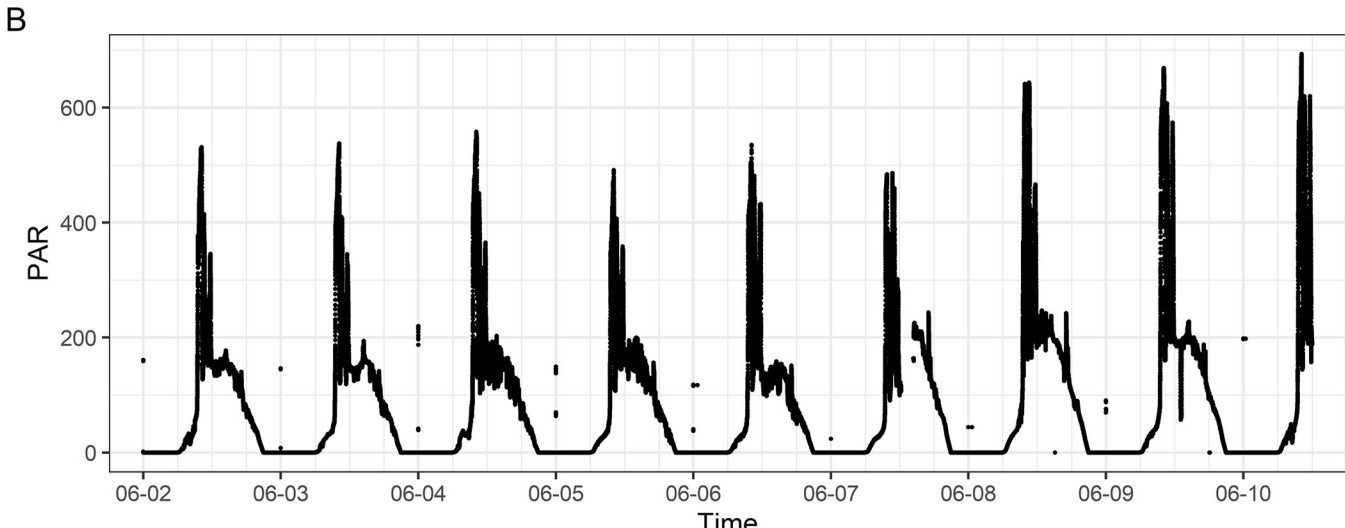

**Fig 2.** A: Valvometric data collected over 8 nights in early June 2021 from Clam 3. Each blue point is an individual observation of percent closure. Red circles delineate intermittent closure events, with the size of the red circle corresponding to the length of the closure event. B: Photosynthetically active radiation (PAR, 400–700 nm) measured by the Li-Cor light sensor hanging over the clams, in μmol photons per m$^2$s. The peaks appearing at midnight are believed to be a programming-related artifact due to a software bug in the LI-1500 light logger, and was confirmed to not be related to an actual flash of light at midnight.

programmed to turn on at 8 AM. The sun rises on the east-facing portion of the Biosphere 2 campus where the clams are located (S2 Fig), leading to the peak in light intensity in the mid-morning hours (Fig 2B). By early afternoon, the measured light levels declined as the sun fell behind the trees overhanging the Biosphere 2 ocean. Clams had reduced incidence of closure events during peak daylight hours (4 AM—2 PM), when light was above ~100 μmol photons/m$^2$s, mostly closing in response to shadowing from a researcher or from the light sensor apparatus hanging above [19]. Clams showed no significant habituation to researcher presence during the experiment, continuing to close when shadowed from above throughout the experiment. Clams began partially closing their valves in the early afternoon around 2 pm as the sun passed behind trees to the west of the Biosphere 2 Ocean and light fell below around 100 μmol photons/m$^2$s. By evening, the clams were generally in a state of persistent partial closure, with a mean closure percentage of 44.1% (SD 31%) in the low-light hours, as compared to 12.7% when light was above 100 μmol photons/m$^2$s. Subsetting times when light was virtually absent (hourly mean PAR< 5 μmol photons/m$^2$s), we found that the clams displayed a statistically significant difference in closure percentage (median 42.8% closure in dark hours compared to 16.6% closure when light was present; Mann-Whitney Wilcoxon Test: W = 5.44 x $10^{11}$, p < 0.001). However, individuals varied in the reliability of this behavior, with some showing more consistent nightly closure than others (Figs 3 and 4A, S3 Fig). Clam 1 showed a mean closure percentage of 29.6% across the experiment (SD 19.6%), Clam 2 showed a mean of 57% closure (SD 29.4%), and Clam 3 showed a mean 42.7% closure (SD 31.5%), suggesting great variation in the closure percentage between and within individuals.

In addition to being more closed between 2 PM and 4 AM, the clams also displayed frequent intermittent rapid valve closure events ("valve-clapping") during that time, as indicated by sudden peaks in the measured voltage data. These events persisted for seconds or minutes in length and were longer in afternoon and nighttime hours when less or no light was present

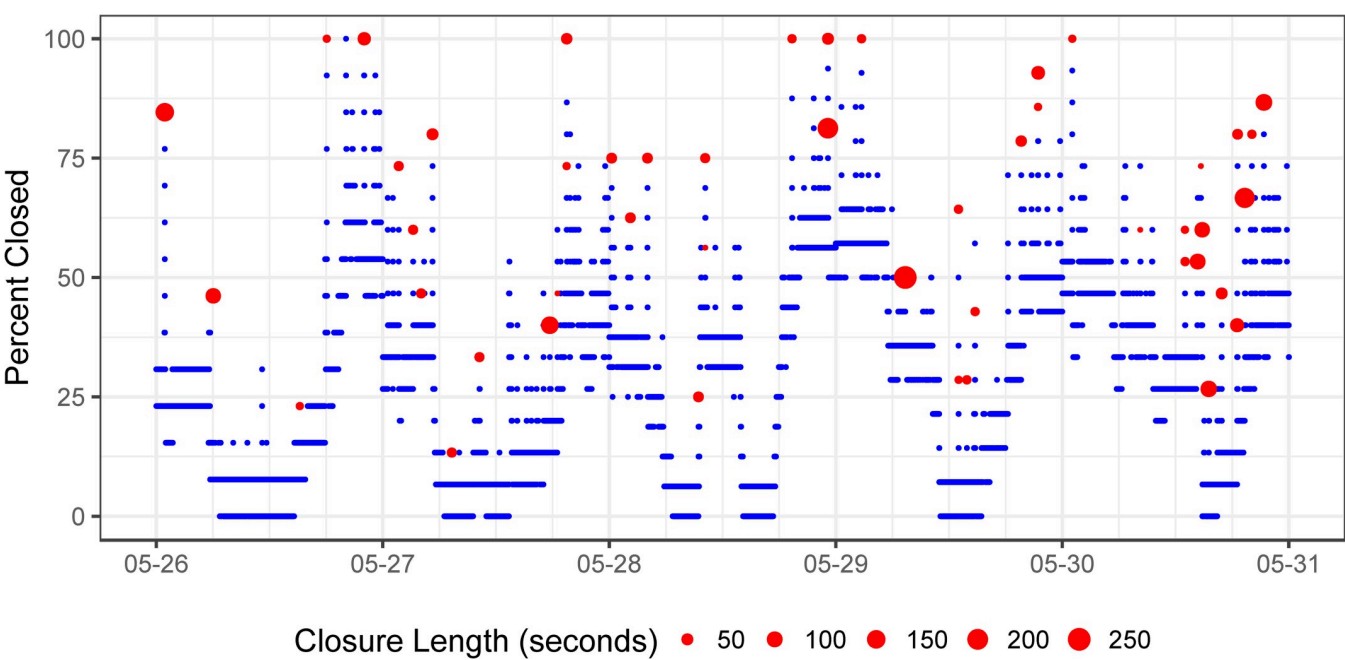

**Fig 3. Data from 6 nights of Clam 1 in late May 2021, with each blue point representing an observation and red circles representing intermittent closure events.** While the day/night pattern is similar to Clam 3 above, there are some daytime extended closure events, such as May 28, which may represent a response to researcher presence by the clam.

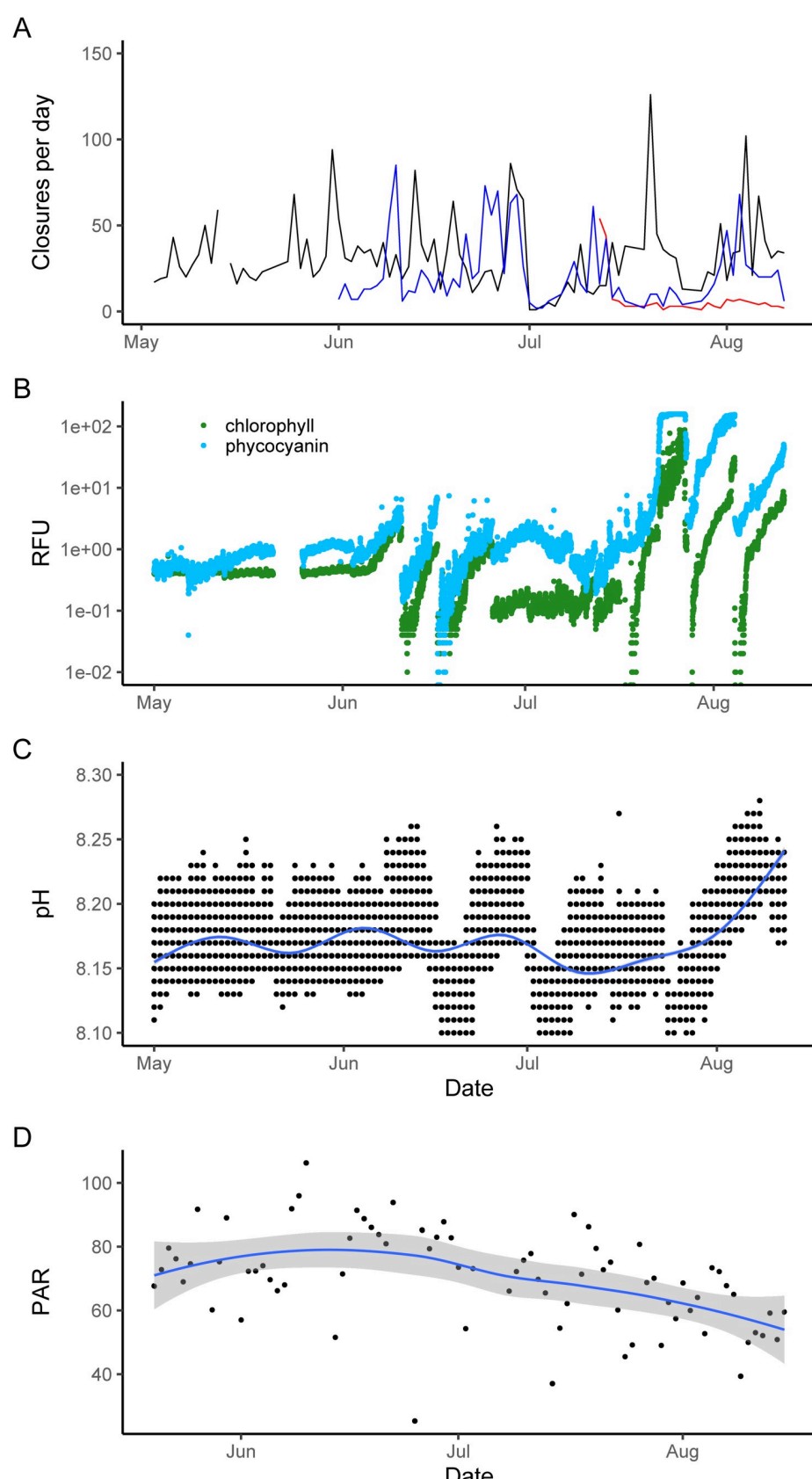

**Fig 4.** A: Closures per day between early June and early August. Different colors are individual monitored clams. B: Log-scaled relative fluorescence units (RFU) for chlorophyll-a (green) and Total Algae Phycoerythrin (blue). C: pH with Loess-fit trendline. D: PAR (μmol photons/m²s.

($< 50$ μmol photons/m²s). In the month of August, closures during the nighttime hours were shorter on average than they were during the daytime, but only Clam 1 showed a statistically significant difference (Mann-Whitney Wilcoxon test: W = 128578, p <0.001), while clam 2 and clam 3 did not reach significance (Clam 2: W = 2031498, p = 0.3; Clam 3: W = 408318, p = 0.089). Between May 3rd and August 10th, there were a mean 34 closures per day, but with great variability around that mean (SD = 27) (Fig 4A).

These valve closing behaviors displayed periodicities related to environmental conditions of the Biosphere 2 Ocean. A generalized additive model showed a significant relationship between daily closure frequency and pH as well as the logarithm of chlorophyll-a, but not light, phycoerythrin, or DO (Table 1, Fig 5). Further, CCF analysis found a statistically significant positive correlation between chlorophyll-a RFUs and valve closure frequency with a lag of 4 days, and a significant 0-day lag with pH (Fig 6).

A wavelet transform of nearly three months (May-August) of minute-scale valvometric data from sensor 3 showed power levels peaking around a 24-hour periodicity. Small amounts of day-to-day variation in the strength of this diurnal cycle are visible throughout the experiment, with one particularly notable multi-day disruption having occurred in late July (Fig 7A). When the 24-hour periodic component of the wavelet was subtracted from the data and a second wavelet analysis was conducted on the residuals, some significant ultradian periodicity was observed at 6 and 12-hour levels (Fig 7B). The wavelet coherence shows high, significant levels of coherence at 24, 12, and 6 hour intervals for light, pH and DO (S2 Fig)); the diurnal variability was in phase with pH and DO, but out of phase with light.

## Discussion

### Dynamics of valve behavior compared to prior studies

The data from the valvometric sensors in this study support the limited prior observations that wild giant clams bask in the daylight hours and then close to around 20% open at night [3], with more valve closures at night than during the day. Similar to the wild-type clams, the Biosphere clams closed to around 30% and experienced more valve closures in the afternoon and nighttime (n = 864 over three months) than during the day (n = 684). This giant clam behavior balances the need to gather photosynthetic energy with the need to avoid predation [16]. Closing the valves actively uses energy through exertion of the adductor muscle, while gaping open

**Table 1. GAM summary.**

|  | Df | Sum squares | Mean squares | F-value | P-value |
|---|---|---|---|---|---|
| S(log(chlor) | 1 | 2081.9 | 2081.89 | 4.4162 | 0.041 |
| S(log(phycoerythrin)) | 1 | 80.8 | 80.77 | 0.1713 | 0.681 |
| S(pH) | 1 | 2673.6 | 2673.58 | 5.6713 | 0.022 |
| S(DO) | 1 | 615.2 | 615.25 | 1.3051 | 0.259 |
| Light | 1 | 274.8 | 274.78 | 0.5829 | 0.44917 |
| Residuals | 45 | 21213.8 | 471.42 |  |  |

Generalized additive model summary of Anova for Parametric Effects, relating daily closure frequency to spline-smoothed chlorophyll-a RFUs, phycoerythrin RFUs, pH and DO. Only pH and logarithm of chlorophyll-a show significant results.

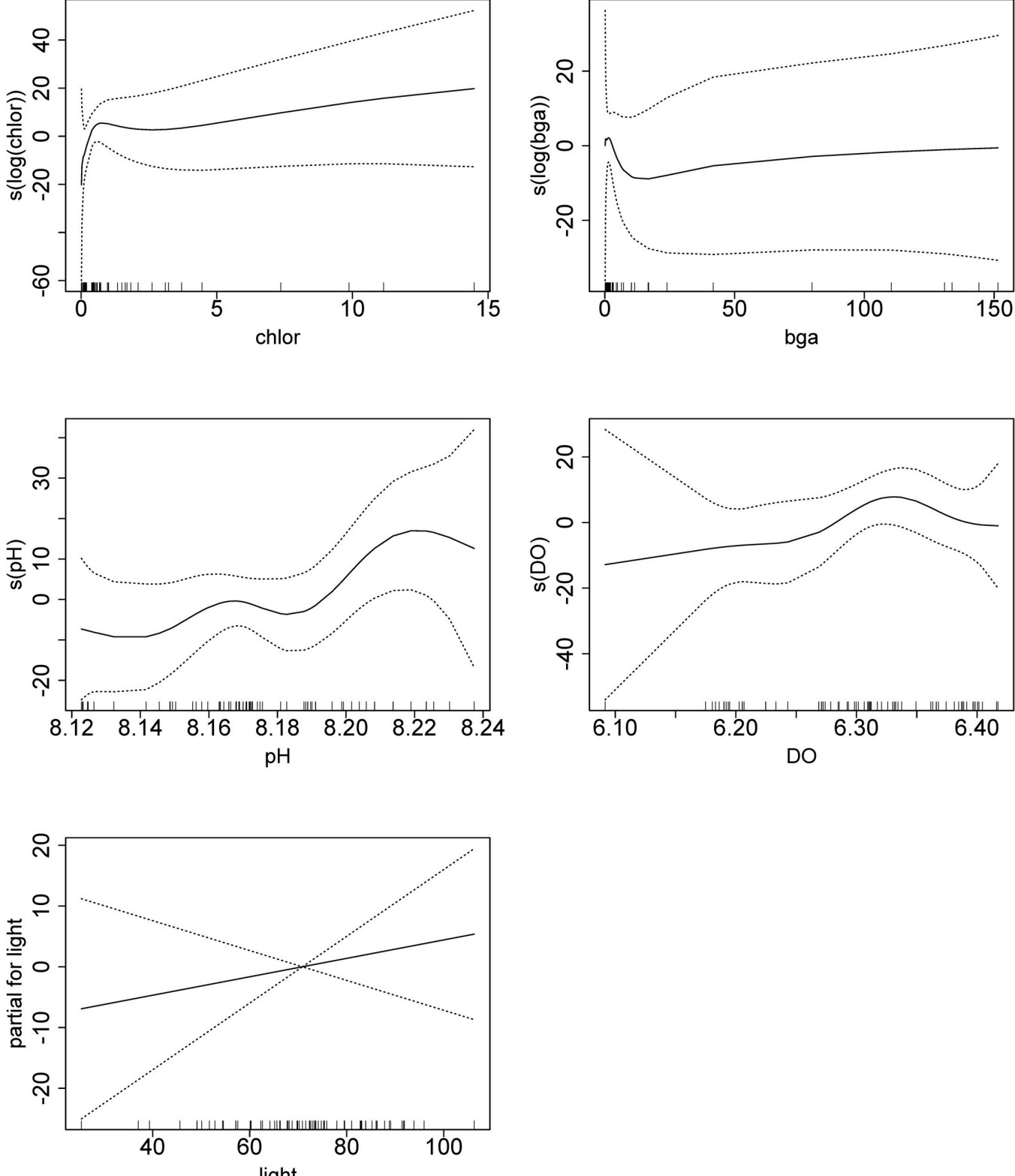

**Fig 5. Generalized additive model (GAM) output showing the influence of the different environmental variables on closure frequency.** All graphs except for light are shown with cubic spline smoothed output of the response variable (closure frequency). The s(x) terminology of the y-axes refers to the output of the pH, DO, chlorophyll-a, and phycocyanin plots having a spline applied. Splines are localized, weighted, additively fitted smoothing functions. They allow for

the analysis of nonlinear relationships of varying distribution which would violate traditional linear regression assumptions [34]. See the introduction for more explanation. Of the variables, only pH and log(chlorophyll-a) show statistical significance (Table 1).

is energetically "free" due to the elastic hinge ligament. Giant clams therefore spend the brightest hours exposing maximum tissue surface area to sunlight for as long as possible. The short closures were likely "valve clapping" events to expel pseudofeces (excess non-food particles adhered to the gills) [37]. We propose that most valve closures at night were related to this filter-feeding activity, based on the average length of those closures. The expelled pseudofeces was visible as "snowdrifts" of fine algal particles surrounding the clams.

Our results contrast with a previous culture experiment exposing *T. maxima* to differing levels of ultraviolet radiation [4], where the clams used partial valve closure to reduce the area exposed to UV at all hours. In the Biosphere 2 Ocean, UV exposure is nearly eliminated by the presence of the glass roof above, reducing these clams' need to utilize such a strategy. More studies are therefore needed into the varying degrees to which valve closure is used as a UV management strategy in wild clams, particularly at differing depths.

The nighttime closures for the Biosphere 2 clams were also more frequent than observed in the prior studies, which may be partially due to the high algal productivity observed in the Biosphere 2 Ocean compared to that of most oligotrophic reefs [27]. Consistent with the expulsion of pseudofeces, most closures were less than 100 seconds in length (Fig 8). By emphasizing filter feeding at night and photosynthesis during the day, the clams effectively gained the nutrition of a phototroph and nighttime heterotroph combined.

The histogram of closure lengths also shows a long-tailed distribution, with some lasting for multiple minutes (Fig 8). These longer events likely represent predator evasion, as the Biosphere 2 Ocean is home to fireworms of the species *Hermodice carunculata* (S4 Fig) [27]. They are slow-moving predators that do not specialize in feeding on bivalves, but are opportunistic in their food choice [38] and were likely attracted to any sessile source of protein available in the Biosphere 2 Ocean. The worms were more active in the evening and at night [38], and were observed during this experiment attempting to enter the clams' shells via the byssal opening. Nevertheless, *T. derasa* has a mostly closed byssal opening [39] unlike other giant clams not adapted to free epifaunal living. By partially closing at intervals throughout the night, the clams may have prevented the worms from attacking through their byssal openings from below. In a previous valvometric study with *Tridacna maxima* in a smaller aquarium tank with no other occupants, the clams were not observed to partially close at night [4], unlike the Biosphere 2 and wild clams. This partial night-time closure may therefore represent a defensive posture in which the clams invest energy when exposed to predation from primarily nocturnal predators such as crabs and fish in the wild, or fireworms in the Biosphere 2 Ocean.

During the day, the average length of closures was around 1.5 times as long as at night, likely in response to potential threats. The primary stimulus during the daylight hours were researchers conducting husbandry activities via snorkeling in the late morning or early afternoon. When giant clams see an object overhead, they will close their valves for a length of time proportional to the novelty and severity of the threat as indicated by object and shadow size [19]. These researcher avoidance events are visible as large, extended closure events in the late mornings and early afternoon (Fig 2A), contributing to the longer daytime closure average. More investigation is needed into whether these results differ for giant clams in a wild environment, where a greater variety of vertebrate and invertebrate predators are present (including octopi, predatory snails, crustaceans, and fish) relative to the Biosphere 2 Ocean (fireworms and researchers).

## CCF: Chl-a and Feeding activity

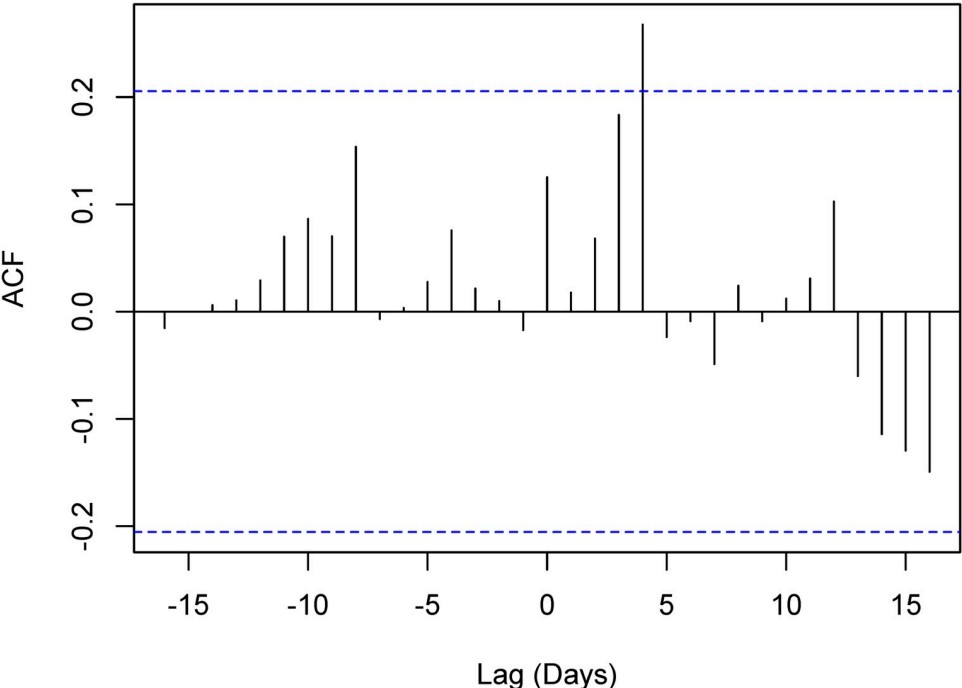

## CCF: pH and Feeding Activity

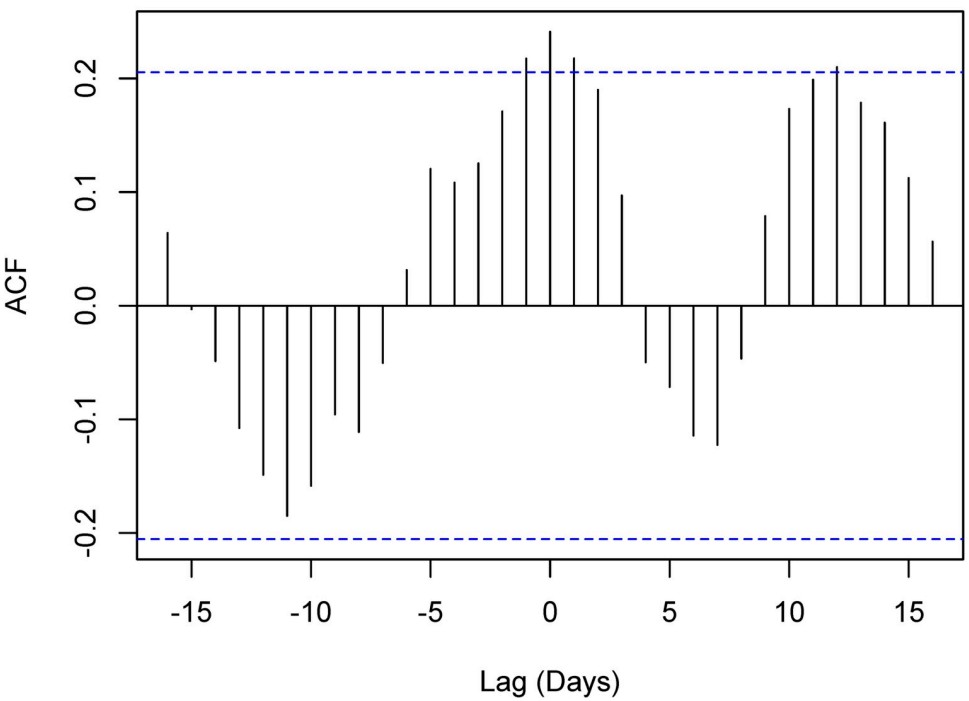

**Fig 6.** A): Cross-correlation function relating daily mean chlorophyll-a RFU values to daily closure frequency. There is a significant positive correlation at a 4-day lag (i.e., closure frequency lagging chlorophyll, as indicated by horizontal blue dashed line). B): Cross-correlation function relating daily mean daily pH values to daily closure frequency. There is a significant positive correlation at a 0-day lag (i.e., in phase).

Over the multi-month experiment, no long-term trend in valvometric activity was observed (Fig 4A), and the clams' behavior was strongly mediated by a 24-hour circadian cycle (Fig 7A). Corroborating past observations of wild clams [3], no significant cyclicity above the 24-hour period was observed (Fig 7A). This contrasts with other bivalves such as oysters which can show fortnightly or lunar day-related periodicity in valve activity [40, 41]. The Biosphere 2 ocean does not have tides, but the fact that prior work in wild tidally-influenced settings also did not identify any lunar cyclicity suggests diurnal light levels are the prime controls on giant clam valve activity. Corroborating the primacy of light as a control on clam behavior, when the

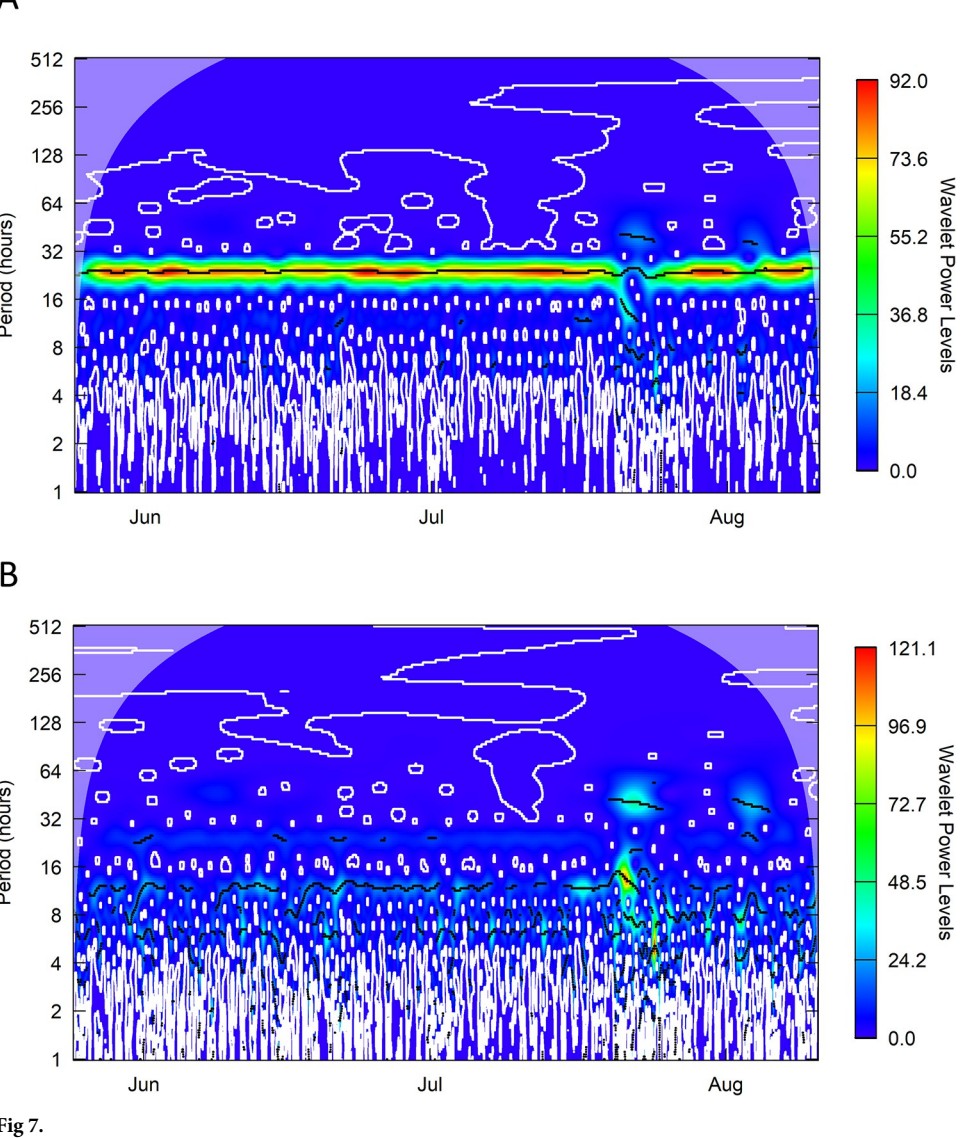

**Fig 7.**

24-hour component of the wavelet reconstruction was removed, a wavelet analysis of the residuals showed some power around 6 and 12-hour periodicities (Fig 7B), which likely relates to the 8 am to 8 pm schedule of the lights above the clams. Some bivalves have been observed to have genetically conserved tidally-mediated valvometric rhythms, even when tides are not present [42]. But such circatidal rhythms were not observed in the prior study of wild giant clams [3]. Wavelet decomposition does not seem to have much prior application to valvometric literature, but these results suggest its broad utility to identify periodic signals in the complex, often noisy valvometric data.

Plotting the coherence of light, pH and DO with valvometric data, we observe high levels of coherence at 24, 12, and 6 hours, likely relating to the diurnal pattern and intradiurnal patterns displayed in each of those parameters (S1 Fig). The arrows in the coherence diagrams indicate pH and DO are in-phase with diurnal valve closures, with higher closure values being seen during times of higher pH and DO (consistent with higher productivity and greater clam feeding). Light, however, is anti-phased, as the clams enter their defensive closed posture at night. Intermittent disruptions in the 24-hour periodicity were observed during the experiment, particularly during one interval in late July. These could be caused by disruptions to the diurnal pattern relating to researcher disturbance leading to prolonged valve closure.

Markers of algal activity in the Biosphere 2 ocean experienced fluctuations during the experiment, including increased mean pH and increased DO indicative of higher algal activity, and repeated spikes in pigment markers of phytoplankton activity during mid-June and late July-early August (Fig 4B and 4C). Daily valve closure frequency was significantly related to ambient pH and log(chlorophyll-a) (Fig 5). The correlation between valve closure frequency and pH was significant on a zero-day lag (Fig 6B). In the Biosphere 2 Ocean, pH varies as a function of photosynthesis from both micro- and macro- algae, which can lead to intermittent bursts of circulating algal debris. At night, respiration draws pH down to lower levels as $CO_2$ is introduced to the seawater, while calcification is not currently occurring at significant enough rates to influence the overall pH of the system.

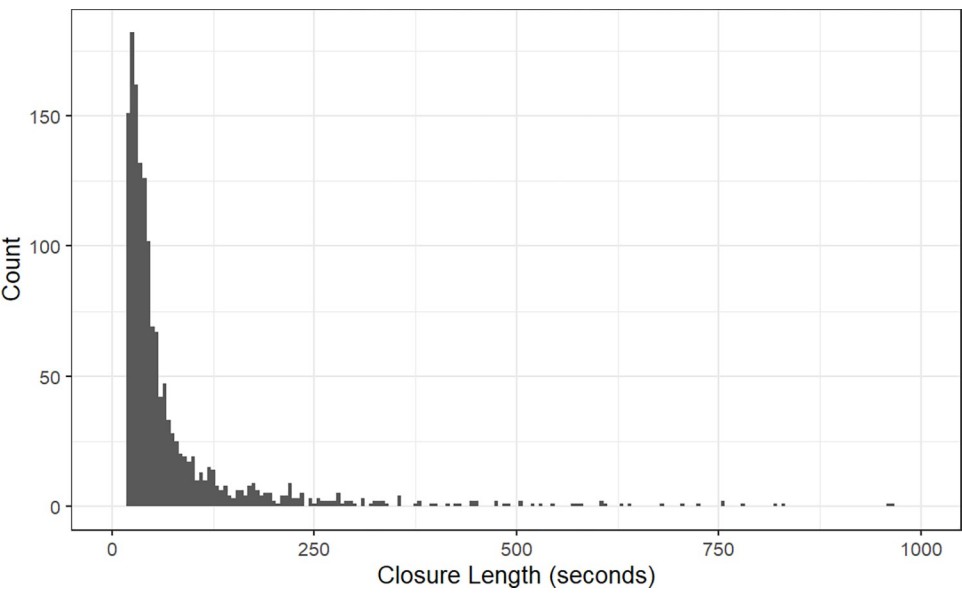

**Fig 8. Histogram of the distribution of closure event lengths for Clam 3 in seconds over the interval from June 1-August 15.**

Prior studies demonstrated that *Tridacna* exposed to sediment show increases in valve closure activity as the clams expel adhered particles from their gills [43]. The Biosphere 2 ocean clams have multiple species of macroalgae growing between and around them on the shallow floor which frequently dislodge and form floating debris that can be captured by the clams' gills. Such detritus has been found to be an important food source for juvenile giant clams [44], but fecal or stomach sampling of the Biosphere 2 clams will be needed to determine the degree to which they are actually digesting such particles. Nevertheless, chlorophyll-a also displayed a significant relationship with closure activity on a four-day lag (with closure frequency lagging chlorophyll-a; Fig 6A), suggesting the clams were capturing circulating phytoplankton in the Biosphere 2 Ocean water, possibly in the periods of die-off following the repeated bloom events that occurred over the summer. Two of the clams showed more synchronized responses than the third (Fig 4A), which did not display the same frequency of valve-clapping, suggesting some amount of inter-individual variability in the behavior.

If the clams compensated for lack of photosynthetic nutrition by increasing filter feeding, then we would expect that closure frequency would be inversely correlated with light exposure. However, because the seasonal light availability over the course of the experiment was small (Fig 4D), mediated by the supplemental lighting rig available at Biosphere 2, we were unable to observe if this was the case. Therefore, no significant relationship between daily mean light level and closure frequency was observed. Further research would benefit from observing high-latitude tridacnids like those of the Northern Red Sea, Okinawa or the southernmost Great Barrier Reef, where changes in insolation on a seasonal basis might be greater.

Additionally, we observed no significant relation between phycoerythrin fluorescence and clam activity. While giant clams, like other bivalves [45], are thought to ingest picoplankton including cyanobacteria [46, 47], this behavior was not visible in the valvometric data in response to phycoerythrin availability, despite significant blooms of cyanobacteria (particularly in August; Fig 4B). It is uncertain whether this was due to a genuine lack of feeding response in the giant clams, or a lack of pseudofecal expulsion of those bacterial particles leading to no recorded "valve-clapping" response.

## Performance of open-source valvometric instrument

The use of open-source components in valvometric research is underreported in the literature. Only a small subset of publications have reported the use of Arduino as a valvometric tool [7, 14, 48], and we could find no mention of Raspberry Pi being used in relation to valvometry.

Purpose-built, specialized marine research microcontrollers might have some benefits in performance compared to these open-source resources, which might explain why they have predominated in past studies. However, we found the performance of this inexpensive system was more than adequate to replicate past valvometric work on giant clams and discern differences in their behavior in the Biosphere 2 ocean. The inexpensive nature of this hardware meant that sensors ($1.67 USD each) and even an Arduino ($20 USD) lost during the prototyping phase of this project were easily replaced. The adhesive putty used for the sensors produced a strong grip but could still be removed from the shell with the pry action of a small screwdriver, minimizing the harm to the clam and allowing for easy replacement of sensors during the prototyping process. The low cost of this hardware puts it in reach for researchers in countries where specialized microcontrollers are more difficult to import, including among the developing countries bordering the Indian and South Pacific oceans where giant clams are widely cultured. Since the initial drafting of this report, pandemic supply chain constraints have led to shortages and price increases for Raspberry Pi hardware [49]. But the open-source nature of the Raspberry Pi platform means that alternative systems such as Orange Pi, Banana

Pi and others can be used in place of the Raspberry Pi system for the same use case of data monitoring and storage. Additionally, the system can be simplified to remove the need for the Raspberry Pi altogether and use the Arduino to log data files directly to SD cards, if a live data view is not needed.

While other studies have configured their systems to operate at a frequency of as high as 10 Hz, much higher than our 5-second frequency [14], we found that the utility of increased resolution was outweighed by the proliferation of data and excessive file sizes. Further, the Hall sensors tended to fail at a higher rate when used at 10 Hz frequency, possibly due to voltage fluctuations from the Arduino. Even at our comparatively slow 5-second sampling rate, weeks of valvometric measurement will accumulate millions of rows of entries and require significant time to process and visualize.

Finally, the sensors as described in this study were reliable over multi-month intervals. The main challenges regarding durability were related to the intrusion of saltwater into the silicone sealant, but these problems were remedied by using aquarium-grade silicone. Where internet connectivity is available, the data can be uploaded continuously via Raspberry Pi to cloud storage for remote access. Alternatively, the Raspberry Pi can be accessed via a keyboard and monitor to transfer stored files to external storage for analysis back at the laboratory. Giant clam valvometry could therefore become a biological complement to the existing instrumentation available at marine stations to understand tropical, coral-reef environments.

## The need for geographically distributed giant clam valvometric data

Another benefit of the open-source hardware is the full reproducibility of the resulting valvometric monitoring equipment. Many research institutions already have Arduinos [25] and Raspberry Pi computers [26] deployed for research projects, and therefore have pre-existing expertise to configure and deploy the valvometric setup as described in our methods. We hope that this could lead to increased use of valvometry in reef environments around the world.

Bivalve biomonitoring has become an increasingly important tool for research into environmental change, but tropical reef bivalves are still under-utilized in this regard compared to their counterparts in temperate climate zones. Other researchers have noted that individuals of the same giant clam species in different regions seem to grow their shells at different times of day [50], have different predation survival rates [51], and different assemblages of symbiont clades [52]. Valvometric data could help quantify how the behavior of these animals influences or is influenced by those observed physiological differences, which in turn could be used to create valvometric metrics for giant clam stress, which are threatened by climate change, over-harvesting and habitat destruction [53]. Valvometric data has already proven useful to understand the response of bivalves to ocean acidification [14] and oil spills [54–56], as well as the impact of their ecological niche on feeding behavior [3, 4]. For example, *Tridacna derasa* has previously been described as adapted to more ocean-influenced reef conditions where turbidity is low, as opposed to *T. gigas*, which has been observed to be more common on inshore reefs influenced by terrigenous productivity [57]. The more extensive valve-clapping in the studied *Tridacna derasa* clams compared to previous investigations of *Hippopus hippopus* [3] and *Tridacna maxima* [4] could therefore be due to variations in pseudofeces expulsion behavior among species. Further valvometric research is needed into whether different giant clam species have varying responses to the same environmental conditions.

Using wild giant clams for valvometry is difficult in most regions due to their protected status. But in a time when giant clams are increasingly cultured for the aquarium trade, food, and their shells [22], there is an abundance of human-bred clams available for valvometric study, many of which are currently being used to re-seed reefs with breeding populations [58].

Valvometers could be deployed among clams at these institutions, using long cables tethered to computing hardware on a nearby shore, at a aquaculture/research station, or in a waterproof housing on a mast. This data could then be readily compared among sites due to the similar open-source resources used, allowing for a distributed dataset of giant clam behavior across the Indo-Pacific. Such a global dataset could help quantify giant clams' response to environmental stress at different latitudes, depths and nutritional regimes, to help delineate where resources must be most urgently allocated to preserve their populations. Because giant clam valve gape patterns are a function of the animal's relative need to obtain photosynthetic nutrition and avoid predation, this data would also be useful to understand the health of their photosymbiotic system and their comparative resilience (to corals and other bivalve species) in the face of the accelerating global climate crisis.

## Conclusion

Valvometry is a useful scientific tool to quantify the behavior of giant clams in a variety of environments. Using a configuration consisting of low-cost and open-source components, giant clam behavior can be monitored in a high-frequency, longitudinal basis. In the Biosphere 2 ocean, the clams entered a posture of partial closure as light levels declined in the afternoon, likely as a defensive strategy against *Hermodice* fireworms present in the ocean tank. During the peak daylight hours, they basked openly and minimized closure time, aided by the reduced predator activity during those hours. This behavior parallels that seen in wild giant clams [3]. Additionally, the clams showed intermittent closure events, mostly at night, which likely represent "valve clapping" to expel pseudofeces. The frequency of these closure events were fairly constant through time, with peaks coinciding with increased pH and lagging the periodic blooms that occurred during the summer in the Biosphere 2 ocean tank. Both pH and excess productivity are dynamics which are currently impacting reef health worldwide, and so giant clams could represent a useful biomonitor for those stressor conditions on wild reefs. More research is needed in settings with greater environmental variability than is present in the Biosphere 2 Ocean. Giant clam valvometry is underutilized but has the potential to identify differences in behavior among species and subpopulations throughout the extensive range inhabited by this unusual subfamily of photosymbiotic bivalves. Giant clams can serve as behavioral sentinels and biosensors recording the present environmental crisis affecting coral reefs worldwide.

## Supporting information

**S1 Fig. Wavelet coherence between valvometric data and light, pH and DO respectively.** In all three diagrams, we see high coherence levels at the 24-hour and 12-hour periodicities, and less so at the 6-hour periodicity. The arrows refer to areas of significant coherence, with right-pointing arrows referring to relationships in phase, while left refers to anti-phased relationships. Light is anti-phased with valvometric activity as the clams record higher closure values at night when light levels are near zero. pH and DO are in phase.
(TIF)

**S2 Fig.** A: Kessil "Tuna Blue" light program, showing the diurnal schedule as % intensity, and approximate PAR levels assuming a peak value of 370 μmoles photons/m$^2$s. B: A view of the floating lighting rig on a cloudy, snowy day.
(TIF)

**S3 Fig. Views from valve sensor output over three nights in late July for clams 1, 2 and 3 from top to bottom, showing individual differences in timing and frequency of nighttime**

**valve closures.**
(TIF)

**S4 Fig. A view of a small *Hermodice carunculata* fireworm, of which there are hundreds of thousands present in the Biosphere 2 ocean.** The smaller individuals frequently attempt to attack the clams through the byssal opening.
(TIF)

**S1 File. Step-by-step instructions on constructing the valvometric sensors.**
(DOCX)

**S1 Data.**
(ZIP)

## Acknowledgments

The authors thank Douglas Cline, who designed and constructed the original valvometric cable prototypes, as well as the floating lighting rig. Wei-Ren Ng assisted with configuration of the Raspberry Pi. Kara Lachapelle and Britney Swiniuch assisted with sensor installation. Franklin Lane assisted with clam husbandry. Lou Law, Engineering Manager at GMW Associates assisted in selection of sensors and sent samples for prototyping. Clams were provided by ORA and Marshall Island Mariculture Farm. Brendan Sullivan and Kessil, Inc. provided lights and troubleshooting expertise. James Fatherree provided advice on clam husbandry.

## Author Contributions

**Conceptualization:** Daniel Killam, Diane Thompson.

**Data curation:** Daniel Killam, Katherine Morgan, Megan Russell.

**Formal analysis:** Daniel Killam, Diane Thompson.

**Funding acquisition:** Diane Thompson.

**Investigation:** Daniel Killam, Diane Thompson, Katherine Morgan, Megan Russell.

**Methodology:** Daniel Killam, Diane Thompson.

**Project administration:** Daniel Killam, Diane Thompson, Katherine Morgan, Megan Russell.

**Resources:** Daniel Killam, Diane Thompson, Katherine Morgan, Megan Russell.

**Software:** Daniel Killam.

**Supervision:** Daniel Killam, Diane Thompson, Katherine Morgan.

**Validation:** Daniel Killam, Diane Thompson.

**Visualization:** Daniel Killam, Diane Thompson.

**Writing – original draft:** Daniel Killam.

**Writing – review & editing:** Daniel Killam, Diane Thompson, Katherine Morgan.

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
