## [Decision Letter · Decision Letter 0]

9 Mar 2022

PONE-D-21-39374Giant clams as open-source, scalable reef environmental biomonitorsPLOS ONE

Dear Dr. Killam,

Thank you for submitting your manuscript to PLOS ONE. After careful consideration, we feel that it has merit but does not fully meet PLOS ONE’s publication criteria as it currently stands. Therefore, we invite you to submit a revised version of the manuscript that addresses the points raised during the review process. We have two reviews from two experts in this field. They both suggest major revisions and offer constructive comments - both state the value in this work and I agree with their conclusions.

We look forward to receiving your revised manuscript.

Kind regards,

David P. Gillikin, Ph.D.

Academic Editor

PLOS ONE

Journal Requirements:

“Funding was provided by the Brown Foundation and the University of Arizona Postdoctoral Fellowship (to DK), and University of Arizona Research, Innovation, and Impact (to DMT)..”

 “DK- Brown Foundation, University of Arizona Postdoctoral Fellowship

DT- University of Arizona RII

Reviewers' comments:

Reviewer's Responses to Questions

**Comments to the Author**

1. Is the manuscript technically sound, and do the data support the conclusions?

Reviewer #1: Partly

Reviewer #2: Partly

2. Has the statistical analysis been performed appropriately and rigorously? 

Reviewer #1: No

Reviewer #2: Yes

3. Have the authors made all data underlying the findings in their manuscript fully available?

Reviewer #1: Yes

Reviewer #2: Yes

4. Is the manuscript presented in an intelligible fashion and written in standard English?

Reviewer #1: Yes

Reviewer #2: Yes

5. Review Comments to the Author

Reviewer #1: In this manuscript, the authors describe a low-cost approach to valveometry for research in fields such as aquaculture and conservation. I am currently conducting valveometry research in my own lab and completely agree with the authors that this is a valuable research approach that is currently plagued by high cost issues. The system they describe seems to be very functional and instructions on how to build it are fairly clear. This part is great. The data they generated seems consistent with the types of data output from higher cost, commercial units (for example, AquaDect MosselMonitors). The authors also do a very nice job of describing some mechanical shortcomings of their system and how to overcome them. These aspects of the manuscript are very useful and will be of strong interest to many readers of this journal.

The other challenge of valveometry revolves around the tremendous amount of data generated, what endpoints to calculate/analyze from this data, and how to define these endpoints. Even something seemingly as simple as “closure” can be defined, calculated, and analyzed in many different ways and it is not always easy to figure out the best way to do this. This aspect of the manuscript is where things seem to fall short and I have many concerns. The system built by the authors has clearly generated a high volume of potentially useful data, but the analytical approaches and logic often seem to be unclear, weak or absent for the conclusions that the authors have drawn. I give several examples below and hope the authors find them useful. These are not easy datasets to analyze and I am sympathetic to the challenges the authors faced.

Introduction

Line 43: Provide subfamily of giant clams.

Line 58: If they are above their maximum thermal tolerance, wouldn’t they be dead and therefore open? Do you mean when they are approaching their maximum thermal tolerance?

Line 59: Doesn’t “The clams close…to minimize exposure to ultraviolet radiation…” contradict the previous statement that they bask wide-open during the daylight hours to optimize photsynthetic production? Daytime is when they would be exposed to UV radiation.

Lines 72-73. Use of electrodes calibrated by a specific formula and connected to a waterproof wireless transmission box would not preclude use of valvometric monitoring in aquaculture. What is the actual problem? The expense of these systems? If so, then say this plainly.

Lines 74-75. “…close-source resources..” Same question why is this more of a problem for the developing world? Expense? Proprietary issues?

Lines 80-84. Why is the use of hot glue or air-curing adhesives considered “invasive”. These are external attachments, not internal (i.e. invasive) attachments. Is the real problem the length of exposure to air during drying rather than ‘invasivness’? Do hot glue or “other air-curing adhesives” require a longer curing time than the Coral Frag Glue used in this study to attach small magnets? It seems to me that this study used very similar, external attachment techniques as previous studies. Did the previous study techniques require an exposure time that was longer than 10 minutes? How much longer? You say these are intertidal animals and can survive hours of exposure, often with shells open. Is reducing air exposure to < 10 minutes really needed for an animal that has evolved to live in an environment with regular, periodic exposure cycles?

Line 91-93. Did both “attachments” cure underwater? I understand the epoxy cured underwater, but did the Coral Frag Glue also cure underwater? How did use of Coral Frag Glue and underwater epoxy allow for easy detachment? Are they weak glues? Weaker than the hot glue and other air-curing adhesives previously referred to? I don’t necessarily doubt the authors when they say their attachments represent an improvement, but strongly suggest more detail to support this claim. I have dealt with very similar problems/issues with adhesives and some information regarding/quantifying the actual improvements in air drying times and ease of removing glued sensors compared to previous adhesives would be very useful.

Methods

Overall, really nice description of sensor methodology and troubleshooting! Much of this section is very clear with a lot of useful information. Just a few concerns

Lines 116-118. So…were the supplementary LED lights strong enough to bring the solar PAR above 200 umol? It is a good idea but did it work? Supplemental Fig 2 (cited in text) suggests it did work for a few hours but the legend suggests this pattern is based on an assumption of a peak value of 370 umoles rather than what was actually achieved. However Fig. 2b provides direct measurements that suggest the LED lights were successful in achieving the goal of >200 umol. Authors should clearly state whether the goal was attained and (I think) cite Fig 2b as evidence.

Lines 191-195. Why use two different glues? Does the Coral Frag Glue produce volatiles? If so, why not just use the underwater epoxy for attachment of the magnets.

Lines 213 – 222. Authors do a nice job of explaining how Z-scores and subsequent Percent closure was calculated, although they say “Percent closure was calculated with the formula:” and the following formula calculates percent open…not percent closure.

Why are results in the figures reported in terms of Zscores and not percent closure? Percent closure is much easier for the reader to understand and interpret. I would strongly suggest reporting % closure rather than Z-scores for most of the figures.

Lines 222-23. Authors need to better explain how “closure events” were defined and calculated. These are very important issues in valveometry and the definition and calculations often differ among studies. Were the authors defining a closure event as a period when % open remained below a specific threshold (i.e. <50%? <10%?), or do they define a closure event as a period of sequential declines in “% open” regardless of what the actual % is (i.e. a decline from 99% to 90% open would be considered a closure event), or a trough within some sort of wave function fit to the % open data? Some other definition? It is not clear how a “..local maxima/minima in a 3-point (>15 second) moving window.” is used to define “closure” and what “closure” actually means. I don’t doubt that they are using a valid technique, they just need to better explain what a closure event is in the context of this study and how it is calculated. This is important because many readers not familiar with valveometry will take “closure event” literally and assume that this means the shell is completely “closed”, especially when the authors say “..in terms of …before reopening.” I do not think a closure event refers to full closing and reopening of the valves in this study.

Results:

Lines 246-261. It is very difficult see the patterns the authors are describing from the figures. Here are some examples.

Line 246: “All three giant clams in this study showed patterns of opening in the hours leading to sunrise, typically around 3-4 Am (Fig 2A).” Figure 2A shows data from only a single clam. No times are provided in Fig 2A, making it very difficult to assess whether this is true.

Line 251 “..reduced incidence of closure during peak daylight hours (4 AM -2 pm), when light was above…”. This time frame does not seem to represent PEAK daylight hours. LED lights did not turn on until 8 A.M. No information is given regarding sunrise, but from what I can tell, the sun doesn’t even rise until ~5:30-7:30 in Arizona, depending on the time of year.

Line 254 “..researcher presence..close when..” here and throughout, it is unclear whether daylight closures in response to shadows from researcher presence is simply speculation or something that was actually measured.

Lines 257-261. Closure percentage means are provided in text but % closure data is not actually shown in the Fig 3 (Z-scores), Fig 4A (# closures), or Supplemental Fig 3 (Z-score). Variance around the means (which was apparently high) is not provided, and there was no statistical analysis testing for differences in % closure between high and low light periods making it very difficult to assess whether mean low-light closures (30%) were really different from mean higher-light closure (12.7%).

Lines 276-285. This section describes “valve clapping” results – frequent rapid valve closure events and compares between lighter and darker time periods. Only one figure (Fig 4A) is referenced here and it describes # closures/day. How closure frequency differs between time periods within a day (i.e. light/dark periods) cannot be discerned from this graph. There is no supporting analysis or data for the statement “..closed once every 34 minutes during darker hours, but only once every 76 minute during lighter hours.” There is weak support for the duration of closures differing between lighter and darker hours (63 vs 90 seconds) in Fig. 2A but this figure is not referenced here and even if it was, it would be difficult to see this pattern. More importantly, as in the previous comment, there is no actual analysis testing whether duration differed between lighter and darker hours, and no indication of the variation around the means of closure duration (63 vs 90 seconds). With regards to the final statement “…no significant long term trends in closures per day were observed…with marked variance around the mean of 34 closures per day (Fig. 4A): Was this data actually analyzed to see if there was a long term trend or did the authors just look at the data and conclude there was not trend. The variance around the mean of 34 closures per day could easily be calculated. What was it?

Lines 286-291. This section is much better. Authors analyzed data using a GAM approach and showed a significant relationship between daily closure frequency and pH as well as log ChlA. They also used a CCF approach to show that valve closure frequency had a lagged response to ChlA but no to pH. My only concern is that it is difficult to interpret Fig 5 because it is not clear what the Y-axis represents. The legend simply says “The s(x) terminology of the y axes refers to the output being smoothed.” It is not clear to me what the Y-axis represents and I suspect it will not be clear to most readers.

Lines 291 – 300. I have to admit, I am totally lost in this section. I had a lot of difficulty interpreting this text and the corresponding graphs.

Discussion.

Given the concerns listed for the Results section, I did not spend a lot of time reviewing the Discussion because it was not always clear which results/conclusions were strongly supported by the data and analyses and which were just speculative. My major suggestion would be to cut the length of the discussion, focusing on pros and cons of this low-cost approach and on the results that are clearly supported by the data. Authors should greatly reduce the speculative discussion and instances of circular logic.

Reviewer #2: This manuscript is well written and deals with the development of low cost equipment to measure the behavior of bivalves by valve spacing. The valve closures and openings show a very clear daily cycle. The statistical treatments here are convincing (cycles at 6, 12 and 24 h, wawelet analysis) It is really interesting. The duration of the experiment is long and this is also an excellent point.

However the results are much clearer on a daily scale than on a monthly scale. I can't understand why

I don't understand how valvometry informs on the resilience of a coral mass (?).

Am I supposed to know Biosphere 2?

Despite the explanations of the authors, I do not understand why the proposed method is non-intrusive since it is a matter of taking a clam out of its environment. I do not understand how it will be made autonomous.

The hall effect sensors can be calibrated since the interval distance is measurable and a range can be achieved at the end of the experiment. The experimentation is done in a giant tank but it seems that the electronics stayed out of the water.

I did not understand the light variations presented in figure 2. The PAR measurement (why a planar sensor?) increases abruptly when you turn on the light. What about the PAR decrease during the rest of the day? Why does the light suddenly increase at midnight? Note that this increase seems to induce a change in the behavior of the clams.

You state that the pH of the sea water is influenced by primary production alone. No. Calcification, respiration, and many other processes influence this parameter.

The legend to your figures is in the body of the text. I do not agree with this inovation.

Do you have proof of the presence of predatory worms Harmodice by video, counting, and direct observation ? .

Closure and opening rhythms could be linked to oxygen concentration in seawater in particlar within the paleal cavity

Translated with www.DeepL.com/Translator (free version)

6. PLOS authors have the option to publish the peer review history of their article (what does this mean?). If published, this will include your full peer review and any attached files.

Reviewer #1: No

Reviewer #2: No

---

## [Author Response · Author response to Decision Letter 0]

30 Apr 2022

Here we present a revised and greatly improved manuscript describing the Biosphere 2 giant clam valvometry work, thanks to the helpful and constructive suggestions provided by reviewers 1 and 2. Reviewer 1’s advice regarding more rigorous statistical analysis on closure rates and lengths has made the manuscript more quantitative and less anecdotal. Reviewer 2’s observations and ideas on additional studies have improved the Discussion section. Both reviewers provided feedback on ways to make the Methods section more transparent and reproducible, through greater elaboration on the rationales behind each methodological choice we made. This is crucial, as this paper’s main purpose is to provide an open valvometric system of use to giant clam workers worldwide. 

The major changes to the manuscript include new figures incorporating the more useful and interpretable “Percent Closed” statistic, instead of Z-scores. We also added aforementioned statistical analyses including pairwise tests and general descriptive statistics to replace our more anecdotal observations in the Results and Discussion. We added more background information on GAM models and wavelet analyses, as they have not previously been much-used in the valvometric literature. And we expanded our description of the reasoning behind our chosen window for delineation of “valve clapping” events. We hope these changes, among other improvements, satisfy the reasonable and helpful advice of the reviewers.

Line by line responses may be viewed in our "Response to Reviewers" word file.

---

## [Editor Report · Decision Letter 1]

23 Nov 2022

Giant clams as open-source, scalable reef environmental biomonitors

PONE-D-21-39374R1

Dear Dr. Killam,

We’re pleased to inform you that your manuscript has been judged scientifically suitable for publication and will be formally accepted for publication once it meets all outstanding technical requirements.

Kind regards,

David P. Gillikin, Ph.D.

Academic Editor

PLOS ONE
---

## [Editor Report · Acceptance letter]

23 Dec 2022

PONE-D-21-39374R1 

Giant clams as open-source, scalable reef environmental biomonitors 

Dear Dr. Killam:

I'm pleased to inform you that your manuscript has been deemed suitable for publication in PLOS ONE. Congratulations! Your manuscript is now with our production department. 

Kind regards, 

on behalf of

Dr David P. Gillikin 

Academic Editor

PLOS ONE